# Interactions between Natural Products—A Review

**DOI:** 10.3390/metabo12121256

**Published:** 2022-12-14

**Authors:** Nemanja Rajčević, Danka Bukvički, Tanja Dodoš, Petar D. Marin

**Affiliations:** Institute of Botany and Botanical Garden “Jevremovac”, Faculty of Biology, University of Belgrade, Studentski trg 16, 11 000 Belgrade, Serbia

**Keywords:** natural products, synergistic effect, therapeutic, extracts, bioactive compounds, in vitro and in vivo studies

## Abstract

Plant-based natural products have been used as a source for therapeutics since the dawn of civilization. According to the World Health Organization (WHO), more than 80% of the world’s population relies on traditional medicine for their primary healthcare. Numerous natural extracts, widely known in Traditional Chinese Medicine, Indian Ayurveda medicine and other practices, have led to the modern discovery and development of new drugs. Plants continuously interact with their environment, producing new compounds and ever-changing combinations of existing ones. Interestingly, some of the compounds have shown lower therapeutic activity in comparison to the extract they were isolated from. These findings suggest that the higher therapeutic activity of the source extract was due to the synergistic effect of several compounds. In other words, the total therapeutic potential of the extract cannot be explained only by the sum of its parts alone. In traditional medicine, most herbal remedies are based on a mixture of plants, and it is the interaction between different constituents that amplifies their therapeutic potential. Considering the significant influence traditional medicine has on human healthcare, knowing and studying the synergistic effect of compounds is paramount in designing smart therapeutic agents.

## 1. Introduction

Plants have formed the basis for traditional medicine systems in many cultures for thousands of years, and traditional medicine will continue to play an important part in health care worldwide. There is enormous potential in natural products, and less than 10% of the world’s biodiversity has been evaluated for biological activity. Phytochemicals with unique structural diversity have long been the primary source of potential drug leads, and many of them have become official drug candidates. Natural constituents and their derivatives have been recognised since ancient times as a source of therapeutic agents and the treatments of ailments, hence becoming an integral part of traditional medicine systems in various parts of the world. Numerous natural constituents (e.g., phenolics, terpenoids, and alkaloids) are attributed with antioxidant, antiseptic, antimicrobial, anti-inflammatory, antiviral, cytotoxic, neuroprotective, and other bioactivities. Due to the wide range of activities, plants and their bioactive molecules are used as natural therapeutics and significantly contribute to the production of commercial drugs. Despite a long history of medicinal use throughout the world, significant utilisation of plants is still limited due to the lack of ethnobotanical information. A combination of natural compounds is usually known to be non-toxic and have synergistic effects. 

Based on their role in metabolism, organic compounds can be divided into two major groups: primary and secondary metabolites. While primary metabolites can be defined as “those molecules that are involved in the biosynthetic pathways of essential components of living cells, such as amino acids in proteins, nucleotides in nucleic acids, sugars as an energy resource and in polysaccharides, or phospholipids as major constituents of cell membranes” [1], secondary metabolites have often been considered those that were not necessary and that were, in essence, by-products of the primary metabolism. However, decades of research into these metabolites [1,2] have shown that, while they have a somewhat limited distribution (i.e., characteristic for specific taxa), they possess a myriad of different specific roles; hence, the term specialised metabolites has been proposed [1,2]. 

While all living organisms produce these metabolites, bacteria, plants, and fungi are the most important. This is probably due to their sedentary lifestyle. While animals can adapt to the environment through behaviour and avoid unfavourable conditions or engage other organisms in many different ways, microorganisms, plants, and fungi cannot. Plants, and to a lesser extent, fungi, are well known for their vast metabolomic diversity [1,3,4,5,6]. No one knows the number of different metabolites produced by plants, but taking into account the estimated number of plant taxa and, at times, massive genomes (especially in polyploid taxa), as well as the ability of some enzymes to produce more than one specialised metabolite [7,8], the number of estimated metabolites may very well exceed 200,000 [1]. 

This review compiles the phytochemicals present in medicinal plants, with a focus on the synergistic effect between extracts and isolated components, as well as their biological properties (in vitro and in vivo). The significance of phytochemicals in anticipation of traditional medicine and their synergistic effects are described herein. Moreover, much work is being done to explore natural products as antimicrobial, anti-inflammatory cytotoxic, antiprotozoal, and antifungal agents. This review gives the highlights of in vitro and in vivo studies of the combination of natural products, especially plants (extracts/essential oils and their individual bioactive components). The aim is to present the interaction between metabolites, whether they originate from a single plant or plant combinations (two or more plants). Beyond herbal combinations, as a part of this review, we went further and gave an interesting overview of different living organisms’ products, for example, plant-plus-fungal extract combinations or bacterial-products-plus-plant-essential-oils, with their medicinal properties, which is a new insight into this evolving topic.

## 2. Natural Products

Specialised metabolites can be divided into several groups based on their chemical nature. The three most numerous groups are terpenoids, alkaloids, and phenolics. There are other groups, but their distribution is fairly narrow–e.g., glucosinolates (Brassicaceae, Capparaceae and some other families), organic disulfides in the Amaryllidaceae family, unusual fatty acids in certain gymnosperms and angiosperms, cyanogenic glucosides in particular members of the Rosaceae family, etc. [1,3,4]. 

Terpenoids are by far the most diverse group of natural compounds, counting over 30,000 compounds (Figure 1). They can be classified based on the number of isoprene units (C5) they possess: Hemiterpenes consist of a single isoprene unit, monoterpenes of two, sesquiterpenes of three, diterpenes of four, triterpenes of six and polyterpenes of seven or more isoprene units. Terpenoids are universally present in all taxa, though sometimes only in small quantities [3,9,10,11]. They play an important role in plant–environment interactions, e.g., attracting pollinators, deterring herbivores, or protecting plants from infections caused by microorganisms [3,12,13,14,15,16,17,18]. They are non-polar compounds, often produced in specialised glands on the surface of plant organs or in resin ducts [12,13,14,19,20]. Most of the studied compounds are part of complex, volatile mixtures called essential oils (EOs) or oleoresins [1,3,9,21]. 

Alkaloids are the second-largest group of natural products, counting over 20,000 different structures (Figure 2). Alkaloids are organic heterocyclic nitrogen compounds soluble in water. The nitrogen in their structure is usually derived from amino acids, though not in all groups. Based on their biosynthetic pathway, they are divided into true alkaloids, protoalkaloids and pseudoalkaloids. Both true alkaloids and protoalkaloids are synthesised from amino acids, although protoalkaloids do not contain heterocyclic nitrogen. Pseudoalkaloids are, however, synthesised differently, e.g., from terpenes or other specialised metabolites. These compounds protect plants from herbivores–they are bitter tasting and oftentimes toxic to mammals and other animals [1,3,9,22,23,24]. While significant, these compounds have limited distribution and can be found only in certain families, (e.g., Solanaceae, Papaveraceae, Cycadaceae etc.) [25].

Phenolics are the third most diverse group of specialised metabolites. They contain over 10,000 different compounds that can be divided into several groups: flavonoids, tannins, and coumarins [1,3,9,26,27] (Figure 3). These compounds differ significantly in their structure and biological role, though they all start their biosynthetic pathway from 4-Coumaroyl CoA. *Flavonoids* are universally present in all plants. They have a 15-carbon skeleton, organised in two phenyl and one heterocyclic oxygen ring. The structural diversity is based on the chemical nature and the position(s) of substituents on different rings, so their polarity varies from more non-polar (aglycones) to more polar (glycosides) compounds. Flavonoids are involved in plant-pollinator interaction (colouring compounds), photoprotection and antimicrobial protection [1,3,6,9,23,28]. *Tannins* are polymeric phenolics found in all plants where they play a role in protection from herbivores and might help regulate plant growth [3,23,29]. *Coumarins* are a group of aromatic benzopyrones consisting of fused benzene and alpha pyrone rings [9,30,31]. Due to their astringent taste, they play a role in protection from predation and pathogenic microorganisms [1,9].

### Medicinal Plants and Phytochemicals

Even before scientists were able to identify different specialised metabolites, plants have been the staple of traditional medicines around the world [32,33,34]. Records of the use of plants in medicine go back about 5000 years, most notably in India and China. Even today, in a modern society dominated by Western medicine, medicinal plants and traditional medicine have their important place. According to the World Health Organization (WHO), more than 80% of the world’s population relies on traditional medicine for their primary healthcare [35]. Numerous natural extracts, widely known in Traditional Chinese Medicine and Ayurveda, led to the modern discovery and development of new drugs. Thousands of years of experience found in ethnobotanical manuscripts, coupled with an ever-evolving scientific approach in the study of these medicinal properties, is a significant opportunity for discovery.

Phytochemicals present in medicinal plants have long been studied [2,36,37]. While having a significant role in plants’ adaptation to the continuously changing environment, these compounds also have a multitude of medicinal properties. Aromatic plants and their essential oils (EOs) have a long history of use in traditional medicine, so it is not surprising that perhaps one of the most studied groups of medicinal phytochemicals is essential oil. Essential oils are complex nonpolar mixtures of volatile organic compounds extracted from all parts of plants, mainly through distillation. These mixtures can have up to 300 different compounds. While in many cases, monoterpenes, sesquiterpenes and volatile diterpenes dominate essential oils, other volatile organic compounds are also found–various carbohydrates, fatty acid derivatives, aldehydes, ketones and esters, among many [7,12,13,15,16,19,20,38,39,40,41,42]. These compounds have various structures–some are acyclic compounds, and others are cyclic or have aromatic rings. The most common compounds include *α*–pinene, *β*–pinene, limonene, *α*–terpinene, *β*–terpinene, *ɣ*–terpinene, *p*-cymene, thymol, carvacrol, pulegone and piperiton as cyclic compounds and linalool, geraniol and citronellol as acyclic. Owing to their larger molecules, even greater structural variability exists among sesquiterpenes (C15). Some of the most common ones are *α*–bisabolene, *β*–caryophyllene, caryophyllene oxide, germacrene D, eugenol etc. [7,12,13,14,15,16,19,21,39,41,43]. The plenitude of literature on the medicinal properties of essential oils is in part due to their abundance, unique composition, and molecular complexity [44,45,46,47,48]. Additionally, other extrinsic factors, like geographical location, phenophase, time of year, sun exposure/orientation, local climate, soil type, and innumerable other factors, can significantly impact the essential oil’s chemical composition [7,13,14,16,19,36,47]. This inherent variability in essential oil composition both benefits plants and influences their medicinal properties. This plethora of different structures means that each compound in this mixture can have a distinct biological property because it will react differently with the substrate. Various medicinal properties have been reported so far for essential oils and their components, e.g., antioxidant, antimicrobial, antifungal, antitumor, anti-inflammatory, analgesic, etc. [40,44,48,49,50,51,52,53,54,55,56,57,58,59,60,61,62,63]. 

Other phytochemicals are obtained by solvent extraction using various methods. These plant extracts may contain more than a hundred individual constituents at varying levels of abundance [33,64,65,66,67,68,69,70,71,72,73,74,75]. However, most of these compounds belong to phenolics, i.e., flavonoids, phenolic acids, tannins, coumarins, non-volatile terpenes (e.g., diterpenes, triterpenes), etc. The obtained extracts’ variation is based on the myriad of different extraction methods and the solvent or solvent combinations used. Thus, different ratios of plant specialised metabolites can be found in the obtained extracts from a single herb. Different medicinal properties have been reported for plant extracts containing phenolics. Flavonoid-rich extracts have shown antibacterial activity, antioxidant properties, anti-inflammatory, antiallergic, vasodilatory, enzyme-inhibitory effects and antitumor activity [9,62,64,65,66,67,68,69,76,77,78]. Phenolic acids are in the focus of current research due to their ability to counteract the consequences of ageing, i.e., the development of cardiovascular diseases, degenerative disorders, and cancer [67]. At the same time, coumarins have shown antimicrobial, antifungal and anti-inflammatory properties [9,77]. A diverse group of compounds, collectively referred to as cannabinoids due to their property to join the cannabinoid receptors of the body and brain, have been extracted from *Cannabis sativa* and other plants (*Echinacea*, *Acmella oleracea*, *Helichrysum umbraculigerum*, *Radula marginata*) [79,80]. These compounds, among many others, exert antitumor properties [79,80]. Medicinal properties of plant extracts rich in non-volatile terpenoids (e.g., carnosic acid, carnosol, cassane, norcassane) have also been investigated. These extracts show antioxidative, antimicrobial, antifungal, antitumor and anti-inflammatory activities [76,81,82,83]. 

Plant extracts may contain a hundred or more individual phytochemicals that vary in their abundance, so the chemical characterisation of these extracts and identification of their components is paramount in understanding their medicinal properties. Until relatively recently, researchers have too often attempted to attribute the medicinal property of an extract to the most abundant compound(s). However, follow-up studies on the medicinal properties of individual compounds extracted and purified have shown either lower activity or, in some cases, even toxicity. Recently, many studies have indicated that the overall activity of extracts is the result of interactions between their components [33]. Indeed, traditional phytotherapy is based on the combination of various medicinal plants [33,34,84,85]. The blend of extracts has proven to have a synergistic effect on their therapeutic properties. In addition, the combination of phytochemicals can reduce the toxicity of individual components in the extract. This effect is reached when components contained in plant extract minimize the negative effects by destroying toxic-acting compound or inhibiting its negative activity and thus providing better activity when added to the original raw drug. Using one of four known methods for preparation of *Radix Aconiti*, the level of toxicity can be reduced to 0.2% [86]. *Rhus hirta* extract, when combined with 5-fluorouracil (chemotherapeutic drug), reduced the toxicity of the drug in vitro, possibly due to presence of antioxidants in the extract [33]. *Juniperus communis* and *Solanum xanthocarpum* when combined in lower doses noticeably reduced hepatotoxicity in vivo studies [34]. The abovementioned interaction is achieved when the compounds affect different target sites or increase the solubility of one another, thus enhancing their bioavailability [33,34,84,85,86]. 

The evaluation of interactions between multiple phytochemicals has gained popularity in recent times. Compounds in extracts can have synergistic, additive, or antagonistic activity [33,87]. Additive and non-interactive combinations indicate that the combined effect of two substances is just a summation effect. Conversely, an antagonistic interaction results in a less-than-additive effect. When the interaction of two or more different compounds or extracts shows a larger positive effect than that of individual components, this positive interaction is considered a synergism. There are several methods deployed in the study of the synergistic effect of compound/extract combinations, though one of the most used are: the general isobole equation [33], fractional inhibitory concentration indices (FICIs) [88], and combination index (CI) [76]. Additionally, several authors have used different statistical methods to assess which compounds from extracts influence different biological activities [60,64,77].

## 3. Plants Extracts vs. Isolated Components and Their Synergistic Effect

### 3.1. Whole Extract vs. Individual Compounds

Initial studies have shown that the essential oils with a higher abundance of certain compounds also show higher bioactivity. However, the activity of the dominant compounds when tested individually was consistently lower or on par with EO/extract (Table 1). For example, the essential oil of four *Mentha* species has shown higher antimicrobial activity connected with the relative abundance of menthol, menthone, piperitenone oxide and carvone [61]. Some seasonal variation in the essential oil composition was also noticed that affected antimicrobial activity, but inconsistently so. In *M. longifolia*, the main compound was piperitenone oxide. The relative abundance of this compound changed from 40% in summer to 64% in winter. However, this did not significantly influence the antimicrobial properties of the oil. The authors also tested the antimicrobial activity of isolated compounds (menthol, menthone, piperitenone oxide and carvone), which was somewhat lower than the total essential oil’s. The essential oil of *Thymus vulgaris* and *Origanum vulgare* was also tested for antioxidant and antimicrobial activity [89]. These essential oils were dominated by thymol and carvacrol, respectively. Both oils showed strong antioxidative and antimicrobial activity, but their respective dominant compounds, when tested individually, showed significantly lower activity [89]. Essential oil of *Pimenta* spp. isolated from leaves and berries showed higher antibiofilm activity in comparison to the dominant compound (eugenol) [90]. The authors concluded that the monoterpene hydrocarbons, which in general exert a less antibacterial effect in comparison to oxygenated compounds, attribute synergistically to dominant compounds and attribute to the higher activity of the whole essential oil. However, the nature of the dominant compound is not the only factor. When studying antimicrobial activities of *Thymus pulegioides* essential oil was dominated by *α*–terpinyl acetate. In almost all cases, essential oil demonstrated higher antimicrobial activity than pure *α*–terpinyl acetate itself [91]. Kerekes et al. [92] studied the antimicrobial effect of three essential oils (*Cinnamomum zeylanicum*, *Origanum majorana*, *Thymus vulgaris*) and their major components (*trans*-cinnamaldehyde, terpinen-4-ol and thymol). Thymol showed the most promising effect in single-species biofilms, while thyme oil demonstrated better antibacterial activity in polymicrobial bacterial cultures. Antifungal properties of essential oil of *Artemisia pedemontana* subsp. *assoana* were assessed [93]. While the essential oil dominated by 1,8-cineole and camphor showed antifungal properties, main compounds tested individually or in combinations did not, attesting to the synergistic effect of minor compounds in the essential oil as well. Hossan et al. [88] tested hexane, ethyl acetate and ethanol extracts of 18 medicinal plants used by the Khyang tribe in Bangladesh for their antimicrobial effects against different pathogenic bacteria, including methicillin-resistant strains. Most of the extracts tested showed antimicrobial properties, even against the methicillin-resistant *Staphylococcus aureus* strain. Cinnamaldehyde, eugenol and gallic acids were also tested for their antimicrobial effect, and in most tested organisms showed lower activity than the extracts. 

A synergistic effect was detected for other medicinal properties as well. For example, essential oil from the flowers of *Agastache rugosa* expressed dose-dependent higher anti-mutagenic activity against the AS52 cell line (Chinese hamster ovary cells) in comparison to any of the three individual components (estragole, limonene and anisaldehyde). Additionally, anisaldehyde was the least abundant compound in the essential oil, but showed the highest anti-mutagenic activity of the three tested compounds [94]. Navel orange essential oil, dominated by D-limonene, showed high antioxidative activity in several in vitro antioxidant tests [103]. When tested against A549 cells (human lung cancer) and 22RV1 cells (human prostate cancer), essential oil showed dose-dependent antiproliferative and apoptosis activity. This activity was several folds higher than the individual components tested (i.e., linalool, *δ*–3-carene, *α*–terpineol, decanal, citral, D-limonene and *α*–pinene) [103]. Similar results were also reported for anti-diabetic, skin-whitening and antioxidative activities of *Cinnamomum zeylanicum* and its main component–*trans*-cinnamaldehyde (81.4%) [95]. Di Martille et al. [50] studied the antitumor effect of *Melaleuca alternifolia* essential oil and its main component–terpinen-4-ol on six melanoma cell lines. Even though terpinen-4-ol exhibited an antitumor effect, the synergistic effect with other compounds in the essential oil was lower. The essential oil of *Eucalyptus camaldulensis* is dominated by *p*-cymene, 1,8-cineole, *α*–pinene [96]. While both EO and individual components showed *α*–amylase inhibition activity, the EO’s was superior, demonstrating synergistic effects of compounds present in the whole EO. Pharmacokinetic interactions occur, for example, between constituents of *Artemisia annua* tea that enables more rapid absorption of artemisinin from plant extract than of the pure drug. Some plant extracts may have an immunomodulatory effect as well as a direct anti-plasmodial effect, while others contain multidrug resistance inhibitors. Some plant constituents are added mainly to attenuate the side effects of others (for example, ginger to prevent nausea) [111]. 

### 3.2. Whole Extract vs. Fractions

Rostro-Atlanis et al. [56] studied *Origanum vulgare* essential oil. This oil was dominated by monoterpene hydrocarbons, namely *o*-cymene and *ɣ*–terpinene. The essential oil was fractionated into five fractions using fractional distillation. The fractions differed both in the overall composition and dominant compounds, namely *o*-cymene, *ɣ*–terpinene, and carvacrol. Monoterpene hydrocarbons dominated the first three fractions, while oxygenated monoterpenes dominated the last two. Two oxygenated monoterpenes (thymol and carvacrol) were absent in the first two fractions. These fractions showed different antioxidative and antimicrobial activity due to the composition. Not surprisingly, the first two fractions demonstrated poor antioxidant activity. Still, the activity of the third fraction, which had at least some amount of thymol and carvacrol, was several folds higher. The highest activity was reported for the last fraction, rich in carvacrol and *β*–caryophyllene. Interestingly, the last two fractions had higher activity than the entire essential oil, owing to a relatively lower abundance of monoterpenes and, possibly, some antagonistic effect of compounds present in the mixture. In a similar study of *Thymus pectinatus* essential oil [112], antioxidative and antimicrobial activity crude essential oil, essential fractions dominated (>80%) by a single compound and individual compounds (thymol, carvacrol, borneol, *p*-cymene) were assessed. As in previous cases, the essential oil showed significantly higher activity than any of its dominant components or fractions dominated by a single component, thus attesting to the synergistic effect of the whole essential oil. 

### 3.3. When 1 + 1 Is Not Equal to 2?

The relationships between components in plant extracts are not always synergistic, and the multitude of individual components in these mixtures make it hard to properly assess the contribution of each of the components. When testing antimicrobial activity of *Nigella sativa* essential oil against *Listeria monocytogenes* strains, EO sometimes displayed higher activity than individual compounds, and sometimes lower, e.g., *N. sativa* EO had higher activity than carvacrol, while *p*-cymene always showed lower activity than the EO [57]. Another study of essential oils of *Curcuma longa*, *C. zedoaria*, *Zingiber officinale*, and *Litsea cubeba* showed high anti-trypanosomal activity and low cytotoxicity compared to standard pharmaceuticals. When individual compounds representing the major components in their EOs were tested, some showed the same or similar anti-trypanosomal activity but higher cytotoxicity. In comparison, others like curlone showed much higher activity than Eos, while evincing significantly lower cytotoxicity at the same time [47]. Hammer et al. [113] tested the antifungal activity of tea tree essential oil (*Melaleuca alternifolia*) and its components. The EO and almost all of the individual components showed antifungal activity. However, antifungal activity was much higher for the whole oil compared to most of the individual components. Terpinen-4-ol, the dominant compound in EO, showed higher activity than EO itself but lower than *α*–terpineol, *α*–pinene, and *β*–pinene (all minor compounds) against *Candida albicans* ATCC10231. When studying the effect of hexane and methanol extracts and their fractions of *Juniperus phoenicea* leaves, Keskes et al. [78] reported that the terpene-rich hexane extract had a higher *α*–amylase inhibition activity and no antioxidative activity. In comparison, flavonoid-rich methanol extract showed higher antioxidant and lower *α*–amylase inhibitory activity. The bioactivity of the fractions of methanol extract was higher, and the individual compound (amentoflavone) was even higher. 

### 3.4. Synergy of Compounds

The initial hypothesis that the dominant compound in a plant extract is responsible for its medicinal properties is being replaced by the synergistic effect hypothesis. An increasing number of studies have suggested that interaction between different components in herbal extracts shows much better activity than any individual component. Barring multivariate statistical analyses (e.g., [60]), one of the most utilised methods of assessing this hypothesis was creating binary and ternary combinations of compounds or fractions and systematically testing their synergistic effects (Table 2). García-García et al. [114] discovered the most active binary (e.g., thymol and carvacrol) and ternary (e.g., thymol, carvacrol, and eugenol) combinations against *Listeria innocua*. Bassolé and Juliani [115] reviewed the antimicrobial properties of mixtures of individual compounds. Some combinations, like thymol/eugenol, carvacrol/eugenol, carvacrol/cymene, and carvacrol/linalool, showed synergistic effects against different bacteria. However, some binary blends displayed antagonistic or pure additive effects (e.g., carvacrol/myrcene or cinnamaldehyde/eugenol, respectively). Several authors [56,89,116] reported the synergistic effect of thymol and carvacrol on their antimicrobial activity. When studying synergistic effects of binary and ternary combinations, the ratios of the compounds is also important. Some showed the highest synergistic effects in 1:1 proportion, e.g., cinnamaldehyde/thymol, thymol/carvacrol or *α*–pinene/limonene. Others exerted the most increased synergistic effects in different proportions, e.g., 1,8-cineole/(+)-limonene in ratio 9:1 or cinnamaldehyde/eugenol in ratio 1:4 and 1:8 [115] (and references cited therein). The combination of compounds belonging to different chemical classes also produces synergistic effects. For example, a combination of 3-caffeoylquinic acid (phenolic acid) and artemisinin (terpenoid) in ratio 1:3 exerts a synergistic antimicrobial effect [116]. Two flavonoids, epigallocatechin gallate and quercetin, as well as quercetin-3-glucoside, punicalagin, ellagic acid, and myricetin in different proportions and combinations, have been shown to exert synergism in their antimicrobial activity against methicillin-resistant *Staphylococcus aureus* [117,118]. Chen et al. [119] studied the cytotoxicity and antihyperglycemic effect of both the main alkaloids from *Rhizoma coptidis* and minor compounds like ferulic acid and choline. Co-administration of the main alkaloids and minor constituents displayed lower cytotoxicity and synergistic anti-hyperglycemic effects.

Carnosic acid and carnosol, two diterpenes from *Rosmarinus officinalis* extract, were tested against colon cancer cells [141]. The authors attributed the bioactivity of the extract to these two compounds and their interaction. To test the synergistic effect, Pérez-Sanchez et al. [81] used these two compounds and two more triterpenes, betulinic acid and ursolic acid in single treatments and in pairwise combinations. Individual components showed a dose-dependent antiproliferative effect which was always lower than that of the entire *R. officinalis* extract. Most of the pairwise combinations between di- and tri-terpenes showed additivity or mild synergy, which would explain the better results of the *R. officinalis* extract. On the other hand, the triterpene combination always brought antagonism between them. Another strong example is the synergism between curcumin and piperine. Curcumin (diferuloylmethane) alone, isolated from *Curcuma longa*, has low oral bioavailability due to glucuronidation in the small intestine. Piperine originated from black pepper (*Piper nigrum*) enhances the bioavailability of curcumin by 2000% in humans, due to an inhibition of this glucuronidation and slowing the gastrointestinal transit [111]. Cai et al. [77] investigated the anti-inflammatory potential of *Ipomoea stolonifera* butanol extract, as well as the individual compounds from the extract (scopoletin, umbelliferone, esculetin, hesperetin and curcumin) and their combinations. The results from in vitro analysis show that esculetin, curcumin and hesperetin were the principal constituents that had synergistic effects when used at the optimal ratio. Additionally, the principal constituents were found to work synergistically in the in vivo mouse ear oedema model at low doses. 

While synergism can be assumed in many cases of plant extracts, it is very hard to prove, since hundreds of compounds can be mixed in innumerable ways. Working with complex mixtures consisting of numerous compounds, it is arduous, if not impossible, to test the synergism between them all. Multivariate statistics can provide a helping tool in this. For example, Ivanov et al. [64] studied cytotoxic, wound healing, antioxidant, antidiabetic, antimicrobial and antibiofilm capacities of various inflorescence extracts of *Salvia nemorosa*. Statistical analysis was performed to assess the correlation between individual compounds in extracts and their medicinal properties. In most of the bioassays, high correlations were found between several of the components and bioactivity, suggesting a synergistic effect between compounds in extract. On the other hand, Rostro-Alanis et al. [56] used the multivariate analysis to find the correlation between the antioxidant and antimicrobial activities with the terpenes in essential oil, namely thymol, carvacrol, *β*–caryophyllene, and *α*–humulene. Similarly, Buriani et al. [60] used principal component analysis (PCA) to study the cytotoxic activity of essential oils from *Pistacia lentiscus*, *P. lentiscus* var. *chia*, *P. terebinthus*, *P. vera*, and *P. integerrima* on human tumor cell lines (human adenocarcinoma cell lines: MCF-7 (breast), 2008 (ovarian), and LoVo (colon)). The multivariate approach highlighted the presence of different cooperating clusters of bioactive phytochemicals, which represent the base for further research. 

Another way to analyse bioactivity synergism is to fractionate the plant extracts and test the obtained fractions for pharmaceutical properties. However, in this approach, some important compounds might be missed. These missed compounds might not have medicinal properties of their own, but in fact facilitate or enable the activity of the active compound. Junio et al. [142] used broth microdilution antimicrobial checkerboard assay to evaluate the synergy of a crude flavonoid-rich extract of *Hydrastis canadensis* in the presence of berberine (alkaloid) at a range of concentrations. The alkaloid berberine can be found in various amounts in the extracts of *H. canadensis*. The crude extract was fractionated and tested with and without the presence of additional berberine. In the fraction that contained three flavonoids that did not possess antimicrobial properties (sideroxylin, 8-desmethyl-sideroxylin, and 6-desmethyl-sideroxylin), a 16-fold decrease of the MICs was observed. Since *Hydrastis canadensis* root extracts are rich in alkaloids and the leaves in flavonoids, the authors concluded that an extract mixing two parts of the plant would be a more effective antimicrobial agent against *S. aureus*.

## 4. Biological Properties: In Vitro and In Vivo Assays

The majority of traditional pharmaceuticals rely on the combination of multiple medicinal plants. In fact, the traditional theory of treatment of diseases is with formulae containing several herbs, in which the medicinal property of one herb is prolonged or enhanced by the other, or the negative effect of one is decreased by the action of the other [85]. In other words, while testing the activity of individual components and their individual interactions is one aspect of synergy study, there is another way–testing synergy of blended plant extracts (Table 3). To validate the traditional use of some plants, in vitro/in vivo bioassays are used.

### 4.1. In Vitro Assays 

In the past decade, much attention has been paid to in vitro antimicrobial activity screening methods. Several bioassays, such as microdilution method, disk-diffusion, well diffusion and broth or agar dilution, are the processes mainly used to prove bioactivity of plant extracts and components isolated. Various specialised plant metabolites from different vascular and nonvascular plants are defense mechanisms against infections caused by pathogenic microorganisms, and their antimicrobial activity has been proved in a number of scientific articles [106,175,176,177,178,179,180,181,182,183]. Plant-based natural products (essential oils, plant extracts and their antimicrobial compounds) found an interesting application in food preservations as natural food antimicrobial preservatives [177,178,180,184,185,186,187,188,189,190]. While this review deals mostly with medicinal properties, the molecular mechanisms that govern antimicrobial activity are universal, and thus conclusions on the medicinal aspect can also be drawn. The antimicrobial properties are based on the different modes of action of specialised metabolites. For example, thymol and carvacrol are small molecules that can easily overcome lipid barriers. Studies on the mechanisms of their action show that they are able to integrate themselves into the lipid layer of the cell membrane, thus increasing the surface curvature. The hydrophilic part of the molecule interacts with the polar part of the membrane, while the hydrophobic part sinks into the membrane. This destabilizes the lipid layer, decreases its elasticity and increases its fluidity, which leads to increased permeability to potassium and hydrogen ions [191]. On the other hand, baicalin, a flavonoid from *Scutellaria amoena*, inhibits *β*–lactamase, an enzyme that catalyses the hydrolysis of penicillins, cephalosporins and other *β*–lactam antibiotics [192]. Alternatively, *α*–pinene, while having low antimicrobial activity, blocks the efflux pump responsible for the ejection of toxic/antibiotic compounds from bacterial cells. The same effect was reported for carnosol, carnosic acid, capsaicin and reserpin [192]. Checkerboard, graphical and time-kill methods are the most widely used procedures to assess the interaction of essential oil components [115]. 

Éva György et al. [146] investigated in vitro antimicrobial properties of different essential oils (thyme, lemongrass, juniper, oregano, sage, fennel, rosemary, mint, rosehips, and dill) and their combinations against some phytopathogenic bacterial strains (*Pseudomonas hibiscicola*, *Brevibacillus agri*, *Enterobacter ludwigii*, *Curtobacterium herbarum*, *Acinetobacter beijerinckii*, *Acinetobacter calcoaceticus*, *Achromobacter xylosoxidans*, *Staphylococcus succinus*, and *Staphylococcus sciuri*). The authors concluded that a synergistic effect was observed in the case of five combinations of essential oils. The most pronounced antimicrobial activity was detected in the case of oregano, while the most promising combination of the essential oil tested was thyme and dill which showed synergism [146].

García-Díeza et al. [148], investigated selected EOs isolated from aromatic and medicinal herbs and spices and their synergistic effect based on antimicrobial activity against food pathogenic bacteria *Salmonella* spp., *Listeria monocytogenes*, *Escherichia coli* and *Staphylococcus aureus*. The authors highlighted that almost all combinations of EOs (thyme, cinnamon, rosemary, cumin, garlic, bay, black pepper, lemon, parsley, and nutmeg) studied displayed a synergic effect against foodborne pathogens. In particular, *Thymus* EO presented the lowest individual MIC, but in combination decreased the MIC of the other EOs. Cinnamon EO also improved the reduction of the individual MICs of the EOs of cumin and parsley. The authors suggest the potential use of EO mixtures to control foodborne pathogens.

Three medicinal plant species (*Merwilla plumbea*, *Hypoxis hemerocallidea*, and *Tulbaghia violacea*) are used in South Africa to treat some infectious diseases [155]. Individual extracts and their combinations were tested in vitro against *Candida albicans* and four bacterial species (2 Gram+ and 2 Gram-). They found that the proportional combination of dichloromethane and petroleum ether extracts of *Merwilla plumbea* bulb showed the strongest synergistic effect against *Staphylococcus aureus*. When studying antimicrobial activity against *S. aureus* and *E. coli*, a combination of emulsified pomelo peel oil and chitosan (polysaccharide) were added in broths of different pH values. Synergistic effects of oil and chitosan (Figure 4) were shown, but only at certain pH [193]. Kachkoul et al. [49] studied the synergistic antimicrobial effect of combinations of three essential oils (*Mentha pulegium*, *Eucalyptus camaldulensis*, and *Rosmarinus officinalis*) against three antibiotic-resistant bacterial strains (*Proteus mirabilis*, *Klebsiella pneumoniae*, and *Staphylococcus aureus*). Individually, the essential oils of *M. pulegium* and *R. officinalis* did not show any activity against *S. aureus*. However, when in combination with another essential oil, the synergistic effect of the combination was recorded.

Combinations of essential oils obtained from *Blepharis cuspidata*, *Boswellia ogadensis*, and *Thymus schimperi* against multidrug-resistant *Escherichia coli*, *Klebsiella pneumoniae* and Methicillin-resistant *Staphylococcus aureus* showed synergistic effects in in vitro tests [147]. In a recent review of antimicrobial activity of essential oil combinations, Cho et al. [125] found that the EO blends showed mainly synergistic or antagonistic effects, rather than additive. Loose et al. [54] studied the antibacterial activity of combinations of essential oils from *Cymbopogon flexuosus*, *Melaleuca alternifolia*, *M. leucadendron* var. *cajuputi*, and *Thymus vulgaris* against different resistant uro-pathogenic species in artificial urine (*Proteus mirabilis*, *Klebsiella pneumoniae* and *Staphylococcus aureus*). By using checkerboard assays, potential synergistic activity combining different essential oils together were determined. While not all blends resulted in synergy, *Melaleuca alternifolia*/*Thymus vulgaris* blend showed an 8-fold increase in antimicrobial activity. Combinations of marjoram and thyme EOs showed antibacterial activities against tested strains. Thus, the lowest MIC of EO combinations of the mixture were *Mentha pulegium* (29.38%), *Eucalyptus camaldulensis* (45.37%) of and *Rosmarinus officinalis* (25.25%) against *P. mirabilis* and combinations of M. *pulegium* (60.61%) and *Rosmarinus officinalis* (39.39%) against *Klebsiella pneumoniae*. Aside from the antibacterial effect, other in vitro studies of the synergy of different extracts combinations have been studied. For example, Parrish et al. [157] tested the combinations of pairs of sub-inhibitory concentrations of selected 65 essential oils against clinical dermatophytes. Combinations with oregano essential oil were synergistic with EOs of cilantro, cassia, and cinnamon bark. Additionally, rose and cassia EOs were found to completely inhibit one dermatophyte. Additionally, antioxidant activities of combined ethyl acetate extract fractions of *Astragalus membranaceus* and *Glycyrrhiza uralensis* showed comparatively better results than any of the individual extracts [159]. 

Pharmacodynamic synergy has been demonstrated between the *Cinchona* alkaloids and between various plant extracts traditionally combined. *Cinchona* alkaloids (quinine and quinidine-isolated from *C. calisaya* and then chemically synthesised) has evolved from bark extracts to chemical synthesis and controlled clinical trials. The molecular mechanisms of the antimalarial activities of cinchona alkaloids are investigated through hemoglobin digestion in digestive vacuoles of plasmodium parasites to hematin. The effects of quinine and quinidine are investigated not only to treat malaria but also to treat atrial fibrillation, restless leg syndrome, Alzheimer’s disease, and epilepsy [194]. However, the use of quinine and quinidine is under a question mark due to their toxicity [194]. 

### 4.2. In Vivo Studies

Apart from in vitro studies which refer to manipulations of organs, tissue or cells under controlled artificial conditions, in vivo studies are often directed for observing the effects of an experiment on a living organism [195]. In spite of the fact that in vivo studies are more expensive and more difficult to control, in many scientific articles, both, in vitro and in vivo experiments are performed. In vivo studies are more suitable for the understanding of the mechanisms of drug-induced toxicity due to their lower structural and functional heterogeneity. For all in vivo methods (antimicrobial, antioxidant, cytotoxic) the natural products that are to be tested are usually administered to the testing animals (mice, rats, etc.) at a definite dosage range. After a determined period of time, the animals are usually sacrificed and blood or tissues are used for the specific assay. 

There are numerous examples of in vivo antimicrobial testing using natural products as active components. Kim et al. [196] investigated the antibacterial activities of 20 natural compounds originating from the plants against pathogenic enteric bacterial strains. The experimental design was made by combinations of in vitro (microdilution and biofilm assays) and in vivo tests (mouse macrophages, reactive oxygen species (ROS) assays). The authors determined a dose-dependent bactericidal and biofilm inhibitory activity of two compounds, honokiol and magnolol, against *Vibrio cholerae*. Combined in vitro and in vivo results suggested that flavonoids and polyphenolic compounds found in food may have protective activity against cholera infection [196].

Kuropakornpong et al. [99], tested a traditional remedy from Thailand consisting of five species: *Piper chaba* Hunt., *Piper sarmentosum* Roxb., *Piper interruptum* Opiz., *Plumbago indica* Linn., and *Zingiber officinale* Roscoe. This study investigated the anti-inflammatory activity of plant extracts and their compounds against PGE2 production in murine macrophage (RAW 264.7) cell line and two in vivo models of anti-inflammatory studies. Extract of this natural product was administered both topically and orally to rats inhibited with inflammation induced by ethyl phenylpropionate (rat ear oedema model) and carrageenan (hind paw oedema model). Results obtained from this work proved traditional remedies and also gave possibilities for the development of phytopharmaceutical products for broader use.

While in vivo studies with natural products are mainly connected to the cytotoxic activity, antiprotozoal, antifungal and other activities are also tested. For example, lavender EO and its components linalool, and linalyl acetate are tested for antitumor activities in vitro on human prostate cancer PC-3 and DU145 cell lines using flow cytometry to study apoptosis induction and cell cycle arrest. For in vivo experiments on mice PC-3 cell line was used to establish subcutaneous xenograft tumors in nude mice. Results confirmed that all three natural products, EO and both compounds showed a stronger inhibitory effect on PC-3 cells than on DU145 cells [105]. Toledo et al., [197] investigated in vitro and in vivo anti-Candida activity of *Cymbopogon nardus* EO in the microemulsion. The experimental in vivo study in a mouse model was performed in female mice. In vivo vulvovaginal candidiasis assay showed that the use of the microemulsion significantly improved the action of the EO.

In vivo antimalarial tests on mice were performed to check the activity of two essential oils (*Cymbopogon citratus* and *Ocimum gratissimum*). At concentrations of 200, 300 and 500 mg/kg of mouse per day, the essential oil of *C. citratus* produced the highest activity with the respective percentages of suppression of parasitaemia: 62.1%, 81.7% and 86.6%. However, before any application of this natural product, toxicity investigation is necessary [198]. Investigation of potential safety and toxicity of natural products are important for in vivo analysis and application for pre-clinical and clinical use of natural products. Dahham et al. [199] tested in vivo toxicity and antitumor activity of essential oils from agarwood (*Aquilaria crassna*). In this study, dose of the EO at 2000 mg/kg/day was given to mice, screened for two weeks, and confirmed as a safe concentration for further preclinical studies. Some in vivo studies, however, proved a therapeutic arsenal against malaria based on traditional medicine. Mota et al. [100] assessed the in vitro activity of EOs obtained from *Vanillosmopsis arborea*, *Lippia sidoides* and *Croton zehntneri*, aromatic plants well known as Brazilian medicinal plants used in ethnomedicine against the human malaria parasite *Plasmodium falciparum*. The toxicity of these oils was assessed in healthy mice and in vitro cytotoxicity was determined at different concentrations against HeLa cells and mice macrophages. The authors investigated individual components of EO (*α*-bisabolol, estragole, and thymol) and concluded that EOs and compounds showed good activity against *P. falciparum* in vitro and in vivo. Another in vivo study showed that the combined lower doses of *Juniperus communis* and *Solanum xanthocarpum* improved the hepatoprotective effect in rats [200]. 

### 4.3. Beyond Herbal Combinations

Combinations of natural products do not end with herbal blends. *Ganoderma lucidum* (fungus) and *Salvia officinalis* have a long history of use in the prevention and treatment of numerous health problems. For example, Ćilerdžić et al. [76] studied the antioxidant and acetylcholinesterase (AChE) and tyrosinase (TYR) inhibition of a fungus-plant ethanol extract. The activity of AChe and TYR is related to the development of neurodegenerative diseases like Alzheimer’s disease. They found strong synergism in antioxidative activity in fungus-plant (ratio 7:3 for dry mixture, 3:7 for extract mixture), and synergism in tyrosinase inhibition activity at 1:1 ratio. Another example of a different origin combination can be found in the combination of water extract of propolis with bee venom in antitumor activity against two breast cancer cell lines, MCF-7 and Hs578T [171]. Drigla et al. found that the synergic effect at higher bee venom concentrations was, respectively, five and two times more pronounced than the two treatments alone. 

As a new generation of antimicrobials, bacteriocins received significant attention, either alone or combined with other natural products (mainly essential oils and their components). An interesting combination occurs between bacterial products such as nisin (bacteriocin produced by a group of Gram-positive bacteria *Lactococcus* and *Streptococcus* sp.) and essential oils. Nissa et al. [152] investigated the antimicrobial activity of nisin and red ginger essential oil (*Zingiber officinale* var. *rubrum*) and their combination against foodborne pathogens and food spoilage microorganisms. The authors concluded that nisin and red ginger EO had a synergistic effect against *Bacillus cereus* and fungicidal effect against *Aspergillus niger* at concentration 62.5 IU/mL of nisin and 1% of the EO. Another group of authors [151] studied a combination of thyme EO at different concentrations (0.3%, 0.6%, or 0.9) and nisin (500 or 1000IU/g), and their combination against *Listeria monocytogenes*. Results showed synergistic activity against the pathogen. The most promising among treatments was the combination of EO at 0.6% and nisin at concentration 1000IU/g. Bag et al. [126], studied synergistic antibacterial and antibiofilm efficacy of nisin and EOs components (linalool and *p*-coumaric) against bacteria *Bacillus cereus* and *Salmonella typhimurium*. The results provide evidence that *p*-coumaric acid with nisin is very effective against biofilms of both bacteria.

Turgis et al. [152] reported the synergistic effect of combined antimicrobial agents on pathogenic bacteria and concluded that combinations of EOs and bacteriocins can act synergistically or additionally to eliminate foodborne pathogens and spoilage bacteria. In the article mentioned, the authors tested six EOs (*Origanum vulgare*, *Cinnamomum cassia*, *Brassica hirta*, *Thymus vulgaris*, *Satureja montana*, and *Cymbopogon nardus*) and four bacteriocins (nisin, pediocin, and two other) against five pathogenic bacteria and two spoilage bacteria.

## 5. Conclusions

Many natural products, which have been widely used as medicinal agents in traditional medicines throughout human history, are now available in commercial supplements and promoted for general health benefits or for prevention and treatment of specific diseases or food supplements, and even additives. Identification of potent herbal mixtures, isolation and identification of compounds and their individual medicinal properties is the first and crucial step for further research aiming at better understanding the underlying mechanisms of their action. This insight, on the other hand, will help us to predict the properties of herbal mixtures and put standards into practice that will ensure the quality control of traditional pharmaceuticals that is needed. This need is being met presently, but the constant rise continues in the number of synergistic studies of different plant extracts. 

Although the search for natural products has intensified, several obstacles still slow down its progress. These obstacles can be found in different areas. Some are in the origin of natural-products-bearing-plants–variation in the quality of natural products could be related to plant variety, agronomic practice, and processing [28,150,195]. Then there are those related to analytical techniques or methods for identifying and characterising the activities of compounds, because botanical extracts contain thousands of individual compounds. It is frequently challenging to assign activity to individual components [101,150,195]. Also, there is a problem with methods of application of natural products, particularly of essential oils; since these compounds have low water solubility, strong organoleptic flavour, and low stability, they require protection from oxygen, light, moisture, and heat. These properties limit their intended usage [51,150,169]. 

The efficacy and safety of products for humans and animals require numerous in vitro and in vivo studies. In vivo studies (which are costly) are often difficult to control. Also, there are ethical issues and extrapolation of data from animals to humans in terms of physiology, biochemistry, genetics, and behaviour. Several EC directives cover food supplements and herbal medicine products that guide medical product identification, such as detailed data on physiological vs. pharmacological and health vs. disease conditions on dose/concentration bases [195]. 

Plant extracts are especially significant in fighting antimicrobial resistance. The uncontrollable use of antibiotics has led to an increase in the number of antibiotic-resistant strains. While sometimes not as effective as an antibiotic, plant extracts still bear some advantages. They are complex mixtures of different natural compounds that affect, kill, or inhibit the growth of bacteria and other pathogenic microorganisms using different mechanisms. Since different mechanisms are responsible for their antimicrobial mode of action, bacteria are less likely to develop resistance to them. Investigations should be carried out both on their mode of action and their cytotoxicity.

One of the main reasons for the isolation and identification of active natural compounds and investigation of their synergistic effect, in the pharmaceutical development process, is the elimination of potentially toxic compounds. Synergy can occur through a variety of mechanisms, for example, through multi-target effects or through modulation of drug transport, permeation, and bioavailability. Synergy can also lead to the elimination of adverse effects, or, in the case of microorganisms, it can circumvent disease resistance mechanisms [33]. Synergism should be investigated at different levels, from in vitro studies, whose role is screening, to in vivo and, consequently, clinical studies of the most promising herbal combinations. Much more clinical research is needed on pharmacodynamic synergy to prove interaction between plant components and also, resistance reversal and attenuation of side effects.

One could expect promising results and solutions from a number of ongoing investigations in many areas (e.g., pharmacy, agronomy, and food science) for which natural preparations are consistent, and clinical studies well-defined. Furthermore, advances in genomics, informatics and associated contemporary ‘omics technologies and their combination with ethnomedical and ethnobotanical studies of traditional medicines will contribute considerably to the speed of discovery, analysis, and development of improved medicines and new drugs [37]. 

## Figures and Tables

**Figure 1 metabolites-12-01256-f001:**
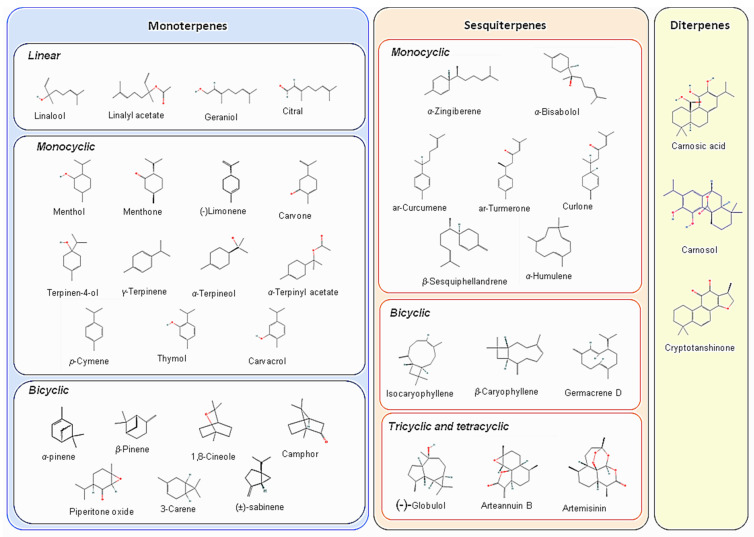
Most-common terpenoid compounds.

**Figure 2 metabolites-12-01256-f002:**
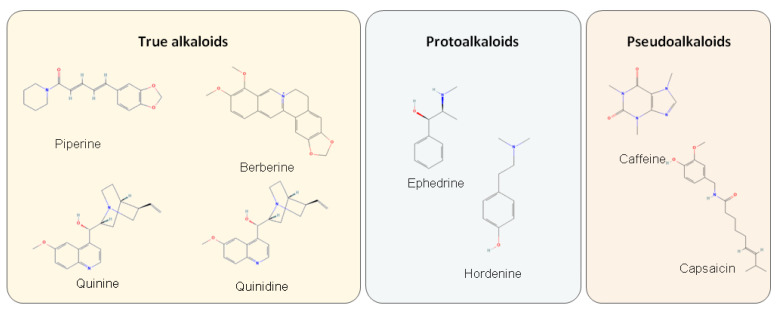
Three groups of alkaloids.

**Figure 3 metabolites-12-01256-f003:**
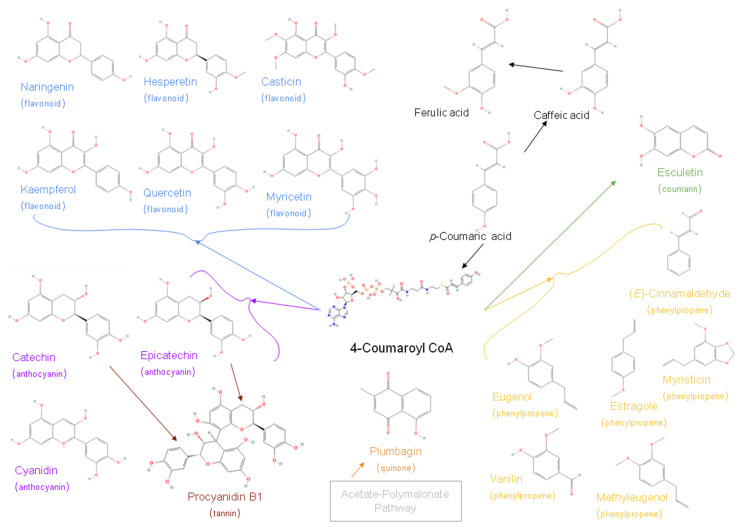
Some of the most common phenolic compounds.

**Figure 4 metabolites-12-01256-f004:**
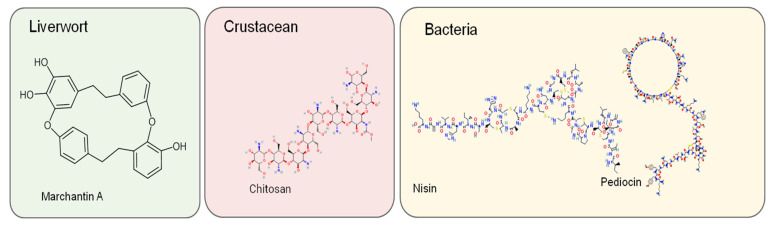
Some of the studied natural products not originating from vascular plants.

**Table 1 metabolites-12-01256-t001:** Interactions between natural compounds in extracts.

Activity	Plant Species	Main Component(s)	Tested Compounds	X *	Ref.
Anti-cancer	*Agasrache rugosa*	Methyl chavicol, Limonene, Anisaldehyde	Methyl chavicol, Limonene, Anisaldehyde	S/I	[94]
Anti-diabetic	*Cinnamomum zeylanicum*	(*E*)-cinnamaldehyde, *(E)*-cinnamyl acetate	(*E*)-Cinnamaldehyde, *(E)*-Cinnamyl acetate	A	[95]
	*Eucalyptus camaldulensis*	*p*-Cymene, 1,8-Cineole	*p*-Cymene, 1,8-Cineole, 1-(S)-*α*–Pinene	I	[96]
				S	
	*Laurus nobilis*	1,8-Cineole, *α*–Pinene	1,8-Cineole, *α*–Pinene, Limonene	S	[97]
Anti- inflammatory	*Bupleurum fruticescens*	β-Caryophyllene, α-Pinene	β-Caryophyllene, *α*-Pinene	S	[98]
	*Piper chaba*	6-shogaol, Piperine, 6-Gingerol	6-shogaol, Piperine, 6-Gingerol	S	[99]
	*Piper interruptum*				
	*Piper sarmentosum*				
	*Plumbago indica*				
	*Zingiber officinale*				
Antimalarial	*Croton zehntneri*	Estragole	Estragole	S	[100]
	*Lippia sidoides*	Thymol	Thymol	A	[100]
	*Vanillosmopsis arborea*	α-Bisabolol	α-Bisabolol	I/A	[100]
Antimicrobial	*Agastache rugosa*	Methyl Chavicol Limonene	Methyl Chavicol, Limonene, Anisaldehyde	S	[94]
	*Angelica keiskei*	Berberine, Magnolol	Berberine, Magnolol, Cryptotanshinone, *α*-Mangostin	A	[101]
	*Caryophyllus aromaticus*	Eugenol (76%)	Eugenol	S/A	[102]
	*Cinnamomum cassia*		Cinnamaldehyde, Eugenol	S	[88]
	*Cinnamomum zeylanicum*	(*E*)-cinnamaldehyde, (*E*)-cinnamyl acetate	(*E*)-cinnamaldehyde, (*E*)-cinnamyl acetate	S/I	[92,95]
	*Citrus sinensis*	D-Limonene	D-Limonene, *α*–Pinene, Linalool, *α*–Terpineol, Citral, 3-Carene, Decanal	S	[103]
	*Croton zehntneri*	Estragole	Estragole	I	[100]
	*Curcuma longa (rhizomes)*	α–Zingiberene, *β*–Sesquiphellandrene, ar-Turmerone	α–Zingiberene, *β*–Sesquiphellandrene, ar-Curcumene, Curlone, ar-Turmerone	S	[47]
	*Eucalyptus globulus*	(+)-Aromadendrene, 1,8-Cineole	(+)-Aromadendrene, (-)-Globulol, 1,8-Cineole	S	[104]
	*Lavender angustifolia*	Linalool, Linalyl acetate	Linalool, Linalyl acetate	S	[105]
	*Lippia sidoides*	Thymol	Thymol	I/A	[100]
	*Marchantia polymorpha*	Terpenes, Oils, Sugars	Marchantin A	A	[106]
	*Melaleuca alternifolia*	Terpinen-4-ol Eucalyptol	Terpinen-4-ol	I	[50]
	*Mentha arvensis*		Eugenol	S	[88]
	*Mentha arvensis*	Menthol	Menthol, Menthone, Carvone	S	[61]
	*Mentha longifolia*	Piperitone oxide	Piperitone oxide	S	[61]
	*Mentha piperita*	Menthone, Menthyl acetate, Limonene,	Menthol, Menthone, Carvone	S	[61]
	*Mentha spicata*	Carvone	Carvone, Menthone	S	[61]
	*Ocimum basilicum*	Estragole	Estragole	S	[102]
	*Origanum majorana*	Terpinen-4-ol	Terpinen-4-ol	S	[92]
	*Origanum vulgare*	Carvacrol	thymol, carvacrol	S	[89]
	*Perilla frutescens*	Perillaldehyde, Limonene	Perillaldehyde, Limonene	S	[107]
	*Pimenta dioica* (berry)	Eugenol	Eugenol	A	[90]
	*Pimenta racemosa* (berry and leaf)	*β*-Myrcene, Eugenol	Eugenol	S	[90]
	*Pistacia lentiscus* var. *chia*	α–Pinene, Myrcene	fractions	S	[108]
	*Salvia hispanica*	Camphor	Camphor	S/A	[102]
	*Satureja hortensis*	Carvacrol	Carvacrol	I/A	[102]
	*Terminalia bellirica*		Gallic acid	S/A	[88]
	*Thapsia villosa*	(*R*)-(+)-Limonene, Methyleugenol	(*R*)-(+)-Limonene, Methyleugenol	S	[109]
	*Thymus pulegioides*	α–Terpinyl acetate	α–Terpinyl acetate	S	[91]
	*Thymus vulgaris*	Thymol, Carvacrol	Thymol	S/I	[89,92,102]
	*Vanillosmopsis arborea*	α-Bisabolol	α-Bisabolol	I	[100]
Anti-trypanosomal	*Curcuma longa (rhizomes)*	α–Zingiberene, *β*–Sesquiphellandrene, ar-Turmerone	α–Zingiberene, *β*–Sesquiphellandrene, ar-Curcumene, Curlone, ar-Turmerone	S	[47]
Immunomodulatory	*Glycyrrhiza* spp.		Glycyrrhizin	S	[110]
	*Hypericum perforatum* (flower)	3-methoxy-2,3-dimethylcyclobutene, *cis-p*-Menth-3-en-1,2-diol	6-methyl-3,5-heptadien-2-one, *β*–Caryophyllene	S	[53]
	(leaves)	Germacrene D, *β*-Caryophyllene, Terpinen-4-ol	Germacrene D, *α*–Humulene, *β*–Caryophyllene	S/A	[53]

* X—Interaction: A—additive, I—indifferent, S—synergistic.

**Table 2 metabolites-12-01256-t002:** Synergism between natural compounds.

Activity	Studied Components	Model/Test	Ratio	Type	Ref.
Anti-inflammatory	esculetin + hesperetin + curcumin	PGE2 release		in vitro	[77]
		NO, IL-6, TNF-a, IL-1*β*, IL-6	12:13:09	in vitro	
		croton oil-induced mouse ear edema	337:191:60	in vivo	
	quercetin + curucumine	COX-2 expression, NFκ*β* activation and NO levels		in vitro	[120]
Antidiabetic	berberine + ferulic acid	HepG2 cells		in vitro	[119]
	chlorogenic acid + ferulic acid	Uptake of 2DG		in vitro	[121]
	(*E*)-cinnamaldehyde + (*E*)-cinnamyl acetate	*α*–Amylase	9:1, 8:2, 7:3, 6:4, 5:5, 4:6, 3:7, 2:8	in vitro	[95]
Antimicrobial	1,8-cineole + camphor	*Aspergillus niger*	1:1	in vitro	[93]
	(+)limonene + (-)limonene	*Cryptococcus neoformans*	1:1		[122,123]
		*Moraxella catarrhalis*			
		*Pseudomonas aeruginosa*			
		*Staphylococcus aureus*			
	artemisinin + 3-caffeoylquinic acid	*Plasmodium falciparum*	1:10-100		[33]
	artemisinin + arteannuin b				
	artemisinin + casticin		1:10-100		
	artemisinin + rosmarinic acid				
	berberine + flavonoid 3,3′-dihydroxy-5,7,4′-trimethoxy-6,8-c-dimethoxyflavone	*Staphylococcus aureus*			[124]
	berberine + piperine	*Staphylococcus aureus*			[124]
	capric acid + thymol, carvacrol, resorcylic acid, eugenol, *trans*-cinnamaldehyde	*Escherichia coli*	1:1		[125]
	caprylic acid + vanillin	*Cronobacter sakazakii, Salmonella typhimurium*	2:3		[125]
	carvacrol + thymol + eugenol	*Listeria innocua*			[114]
	carvacrol + thymol	*Listeria innocua*			
	carvacrol + cymene, eugenol, linalool	*Bacillus cereus,*		in vitro	[115]
		*Escherichia coli,*			
		*Listeria monocytogenes*			
	cinnamaldehyde + carvacrol	*Salmonella typhimurium,*	1:1, 1:2	in vitro	[115,125]
		*Escherichia coli*			
	cinnamaldehyde + thymol	*Escherichia coli,*	1:1, 1:2	in vitro	[115,125]
		*Salmonella typhinurium*			
	eugenol + linalool, menthol	*Enterobacter aerogenes, Escherichia coli,*		in vitro	[115]
		*Pseudomonas aeruginosa*			
	lauric acid + resorcylic acid, carvacrol, thymol	*Escherichia coli*	1:2		[125]
	limonene + 1,8-cineole	*Pseudomonas aeruginosa, Staphylococcus aureus*		in vitro	[115]
	menthol + geraniol, thymol	*Bacillus cereus,*		in vitro	[115]
		*Staphylococcus aureus*			
	nisin + linalool	*Bacillus cereus*		in vitro	[126]
	nisin + *p*-coumaric acid	*Salmonella typhimurium*		in vitro	[126]
	polygodial + perillaldehyde	*Bacillus subtilis,*			[106]
		*Candida albicans,*			
		*Mucor mucedo,*			
		*Pseudomonas aeruginosa, Penicillium chrysogenum, Saccharomyces cerevisiae, Salmonella choleraesuis*			
	thymol + carvacrol, eugenol	*Escherichia coli,*	1:1	in vitro	[115,125]
		*Salmonella typhinurium*			
	gallate + quercetin	*MethicillinResistant Staphylococcus aureus* (MRSA)	1:1, 2:1	in vitro	[117]
	quercetin3-glucoside + myricetin	*Staphylococcus aureus*	3:1, 1:3, 1:7	in vitro	[118]
	quercetin3-glucoside + punicalagin	*Staphylococcus aureus*	1:3, 1:7	in vitro	
	myricetin + punicalagin	*Staphylococcus aureus*	1:3, 1:7	in vitro	
	ellagic acid + punicalagin	*Staphylococcus aureus*	7:1, 3:1, 1:3, 1:7	in vitro	
	ellagic acid + quercetin3-glucoside	*Staphylococcus aureus*	3:1, 1:3, 1:7	in vitro	
Antiproliferative	carnosic acid (CA) carnosol (CAR), betulinic acid (BA) and ursolic acid (UA), combinations: CA+CAR, CA+BA, CA+UA, CAR+UA, and CAR+BA	HT-29 cells		in vitro	[81]
Anti-neurodegenerative	(*E*)-cinnamaldehyde + (*E*)-cinnamyl acetate	Tyrosinase inhibition assay	9:1, 8:2, 7:3, 6:4, 5:5, 4:6, 3:7, 2:8	in vitro	[95]
		AChE inhibition	1:9	in vitro	
		Inhibition of Aβ1-42 aggregation	8:2, 7:3, 6:4, 2:8, 1:9	in vitro	
	1,8-cineole + *α*–pinene, 1,8-cineole + caryophyllene oxide, 1,8-cineole + camphor	Inhibition of AChE	1:10	in vitro	[127]
Antioxidative	(*E*)-cinnamaldehyde + (*E*)-cinnamyl acetate	Phosphomolybdenum, FRAP, CUPRAC	9:1, 8:2, 7:3, 6:4, 5:5, 4:6, 3:7, 2:8, 1:9	in vitro	[95]
	malvidin-3-o-glucoside + pelargonidin-3-o-glucoside, catechin, epicatechin, myricetin, quercetin, quercetin-3-glucoside	FRAP	1:1	in vitro	[128]
	pelargonidin-3-o-glucoside + epicatechin, myricetin, kaempferol, quercetin, quercetin-3-glucoside	FRAP	1:1	in vitro	[128]
	catechin + myricetin, quercetin, quercetin-3-glucoside	FRAP	1:1	in vitro	[128]
	epicatechin + myricetin, quercetin, quercetin-3-b-glucoside	FRAP	1:1	in vitro	[128]
	rosmarinic acid + quercetin, caffeic acid	AAPH-induced oxidation	0-4:0-1:0-5	in vitro	[129]
	quercetin + rutin, catechin, *p*-coumaric acid, cyanidin	Liposome oxidation test, Inhibition of platelet function, ORAC	0.5-1:0.25-0.5, 5:25, 1:1	in vitro	[130]
	*p*-coumaric acid + catechin	ORAC	1:1	in vitro	[131]
	kaempferol + myricetin, quercetin, quercetin-3-glucoside	DPPH, FRAP	1:1	in vitro	[128]
	hesperidin + chlorogenic acid, myricetin, naringenin	ORAC	1:1	in vitro	[132]
Antitumor/anticancer	Cannabidiol + Cannabigerol	leukemia (CEM)HL60, breast MCF-7	1:1	in vitro	[133,134]
	Cannabidiol + Δ9-tetrahydrocannabinol	acute lymphocytic leukemia (CEM)HL60, glioblastoma cell lines U251, SF26	1:1	in vitro	[133,135]
	Quercetin + Curcumin	Chronic myeloid leukemia cell line K562, breast cancer, ovarian cancer		in vitro	[136,137,138]
	berberine + d-limonene	human gastric carcinoma cell line MGC803		in vitro	[139]
	β–caryophyllene + α–humulene, isocaryophyllene	human breast adenocarcinoma cell line MCF-7	1:3	in vitro	[140]

**Table 3 metabolites-12-01256-t003:** Synergistic effect of plant extract with other natural compounds or extracts.

Activity	Botanical Extract	Plant Species/Natural Product Combination	Ratio	Type	Ref.
Anti-Inflammatory	Extract + Extract	*Astragalus membranaceus + Rehmannia glutinosa*	2:1	in vitro	[143]
		*Boswellia carterii + Commiphora myrrha*		in vivo	[144]
		*Coptis chinensis + Phellodendron amurense*		in vivo	[145]
		*Piper chaba* + *P. sarmentosum* + *P. interruptum* + *Plumbago indica + Zingiber officinale*	n/a	in vitro	[99]
Antibacterial	Essential oil + essential oil	*Anethum graveolens + Foeniculum/Salvia/Rosmarinus/Thymus*	n/a	in vitro	[146]
		*Aniba rosaeodora, Thymus vulgaris*	n/a	in vitro	[84]
		*Blepharis ogadensis + Blepharis cuspidata*	1:1	in vitro	[147]
		*Cinnamomum zeylanicum* + *Syzygium aromaticum*	n/a	in vitro	[84,115]
		*Cinnamomum zeylanicum + Petroselimum sativum*	n/a	in vitro	[148]
		*Cuminun cyminum + Cinnamomum zeylanicum*	n/a	in vitro	
		*Cuminum cyminum* + *Coriandrum sativum*	n/a	in vitro	[84]
		*Cymbopogon + Juniperus/Foeniculum/Rosa/Rosmarinus/Salvia*	n/a	in vitro	[146]
		*Cymbopogon citratus + Cymbopogon giganteus*	2:1	in vitro	[115]
		*Eucalyptus camaldulensis + Mentha pulegium + Rosmarinus officinalis*	3:4:2	in vitro	[49]
		*Garlic + Bay*	n/a	in vitro	[148]
		*Juniperus + Foeniculum/Mentha/Rosmarinus/Salvia*	n/a	in vitro	[146]
		*Lavandula angustifolia + Cinnamomum zeylanicum*/*Daucus carota*/*Juniperus virginiana*/*Thymus vulgaris*	n/a	in vitro	[84]
		*Lippia multiflora + Mentha piperita, Origanum basilicum*	16:1, 5:3, 8:1	in vitro	[115]
		*Melissa officinalis* + *Thymus vulgaris*	n/a	in vitro	[84]
		*Mentha piperita* + *Ocimum basilicum*	n/a	in vitro	[84]
		*Mentha pulegium + Rosmarinus officinalis*	06:04	in vitro	[49]
		*Ocimum basilicum* + *Citrus bergamia*	n/a	in vitro	[84]
		*Origanum vulgare* + *Citrus bergamia*, *Ocimum basilicum, Rosmarinus officinalis*	1:16, 1:8	in vitro	[84,115]
		*Salvia + Rosmarinus, Foeniculum, Mentha, Rosa*	n/a	in vitro	[146]
		*Satureja hortensis + Origanum vulgare* subsp. *hirtum*	2:1	in vivo	[149]
		*Thymus schimper + Blepharis cuspidata, B. ogadensis, Melaleuca alternifolia, Pimpinella anisum*	1:1	in vitro	[54,84,147]
		*Thymus capitatus + Cinnamomum zeylanicum*	n/a	in vitro	[148]
		*Thymus capitatus + Cuminun cyminum*	n/a	in vitro	
		*Thymus capitatus + Garlic*	n/a	in vitro	
		*Thymus capitatus + Petroselimum sativum*	n/a	in vitro	
		*Thymus capitatus + Rosmarinus officinalis*	n/a	in vitro	
	Essential oil + Essential oil fractions	*Anethum graveolens*	n/a	in vitro	[150]
		*Coriandrum*	n/a	in vitro	[150]
		*Coriandrum (F9) + Eucalyptus (F2)*	n/a	in vitro	[150]
		*Eucalyptus*	n/a	in vitro	[150]
		*Pistacia lentiscus* var. *chia*	n/a	in vitro	[108]
		*Thymus vulgaris*	n/a	in vitro	[151]
	EO + MT104b	*Cinnamomum cassia*	n/a	in vitro	
	EO + nisin	*Origanum vulgare*	n/a	in vitro	[152]
		*Thymus vulgaris*	n/a	in vitro	[152]
		*Zingiber officinale* var. *rubrum/nisin*	n/a	in vitro	[153]
	EO + pediocin	*Satureja montana*	n/a	in vitro	[152]
	Extract + Extract	*Elephantorrhiza elephantina + Pentanisia prunelloides*		in vitro	[154]
		*Hypoxis hemerocallidea* (different plant organs)	n/a	in vitro	[155]
		*Merwilla plumbea* (different plant organs)	n/a	in vitro	
		*Tulbaghia violacea* (different plant organs)	n/a	in vitro	
	Extract + berberine	*Hydrastis canadensis*	n/a	in vitro	[142]
Antifungal	Essential oil + Essential oil	*Cymbopogon martini + Chenopodium ambrosioides*	1:1	in vitro	[156]
				in vivo	
		*Lavandula angustifolia + Andropogon muricatus, Angelica archangelica, Artemisia dracunculus, Canarium luzonicum, Carum carvi, Citrus aurantium, C. grandis, C. sinensis, C. medica limonum, Cinnamomum zeylanicum, Commiphora myrrha, Cupressus sempervirens, Cymbopogon nardus/Daucus carota, Eucalyptus globulus, Foeniculum dulce, Hyssopus officinalis, Juniperus virginiana, Litsea cubeba, Melaleuca alternifolia, Myrtus communis, Origanum majorana, Pinus sylvestris, Piper nigrum, Pogostemon patchouli, Rosmarinus officinalis, Santalum album, Styrax benzoin, Tagetes patula*	n/a	in vitro	[84]
		*Origanum + Coriandrum sativum, Cassia, Cinnamum*	1:1, 4:1, 2:1	in vitro	[157]
		*Rosa* + *Cassia*	4:1	in vitro	[157]
		*Salvia officinalis + Thymus vulgaris*	n/a		[58]
		*Syzygium aromaticum + Brassica nigra*	9:2	in vitro	[158]
			10:1	in vivo	
		*Syzygium aromaticum + Rosmarinus officinalis*	1:5, 1:7, 1:9	in vitro	[115]
	Essential oil + Fraction	*Coriandrum sativum*	n/a	in vitro	[150]
		*Eucalyptus*	n/a	in vitro	
Antioxidative	Extract + Extract	*Astragalus membranaceus + Glycyrrhiza uralensis*	01:01	in vitro	[159]
		*Camellia sinensis, Cinnamomum cassia, Ginkgo biloba, Phyllanthus emblica, Punica granatum, Vitis vinifera*	5:3:3:3:3:3	in vitro	[160]
		*Salvia officinalis + Ganoderma* (fungus)	7:3	in vitro	[76]
		*Rhus hirta + Rubus strigosus*	1:1	in vitro	[161]
	Essential oil + Essential oil	*Apium graveolens + Thymus vulgaris + Coriandrum sativum*	6:2:2	in vitro	[162]
		*Coriandrum sativum + Cuminum cyminum*	1:1	in vitro	[163]
		*Zingiber officinale + Cinnamomum verum + Elettaria cardamomum*	1:7:2	in vitro	[164]
	Essential oil + natural compound	*Santalum sp + α*-santalol	10:1	in vivo	[165]
Antimalarial	Extract + Extract	*Mitragyna inermis + Feretia apodanthera, Guiera senegalensis*		in vitro	[144]
		*Nauclea latifolia + Feretia apodanthera, Guiera senegalensis, Mitragyna inermis*			
		*Lawsonia inermis + Tithonia diversifolia*	1:1	in vitro	[166]
				in vivo	
Antitumor/Anticancer	Extract + Quercetin	*Lycopodium clavatum*	10 µL: 50 µM	in vitro	[167]
	Extract + Extract	*Coptis chinensis + Evodia rutaecarpa*	6:1	in vitro	[168]
				in vivo	
		*Corydalis + Curucuma*		in vitro	[169]
		*Curcuma longa + Rosmarinus officinalis*		in vivo	[170]
		propolis + bee venom	7:5	in vitro	[171]
		*Vigna radiata + Vigna unguiculata. subsp. unguiculata + Sauropus androgynus*		in vitro	[172]
Cytotoxicity	EO + EO	*Cymbopogon citratus + Cymbopogon giganteus*		in vitro	[173]
		*Salvia officinalis* + *Thymus vulgaris*	n/a	in vitro	[58]
Anti-neurodegenerative	Extract + Extract	*Polygala tenuifolia + Panax ginseng + Poria cocos + Acorus tatarinowii*	3:2:3:2	in vivo	[174]
		*Salvia officinalis + Ganoderma*	7:3	in vitro	[76]

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
