# Peer review of "Interactions between Natural Products—A Review"

_metabolites, 2022, doi:10.3390/metabo12121256_

Round 1

Reviewer 1 Report

The manuscript entitled: “Interactions between natural products – a review is an interesting review with 201 references update to 2022, describes interactions, mostly synergistic effects of plant extracts/phytochemical compounds. In my opinion, the manuscript’s topic is sound. The article has been written well, however needs major revisions, here are some remarks that have to be consider in further submission.   

1.      In the table 1-3, it’s better to arrange the activity and plant species in each activity in alphabet order. It’s important to arrange the summary of research in the table so the readers can follow easily.

2.      Figure 1 is not clear. It’s better to re-draw the structures.

3.      Many interaction references that authors described in the manuscript are not in the table. In my opinion, the table should provide full interaction examples.

Part 2.1. References 88, 90, 111-112 are not in the table 1.

Line 214-215. Reference 93 is not correct. Reference 93 is in the table 2.

Part 2.4. Reference 81, 115, 119, 120, 143 are not in the table 2.

Line 346. Is the Reference 112  correct ? It mentioned about antimalarial activity.

Line 372-373: the sentence “One of the drawbacks of
this approach is that it may not facilitate the identification of compounds that, on their
own, do not possess the medicinal properties, but in fact facilitate or enable the activity
of the active compound” need to rephrased. It’s difficult to understand.

Part3.1. Reference 193, 126, 54 are not in the table 3.

Part3.2. Reference 97, 99, 197-201 are not in the table 3.  In the table 3, there are several in vivo studies but the authors did not describe in detailed any examples.  

Figure 4. Marchatin A is not mentioned in the manuscript. It can be removed.

Part3.3. The authors can make another table for this part, not included in the table 2.

Reference 151. The author omitted nisin in the combination.

4.      Many references in the tables are not described. I understand the manuscript topic is very broad, the authors can not describe in detailed all the references but it seems the authors give many examples about interactions of extracts/compounds in the antimicrobial activity. It’s better to discuss about interactions of extracts/compounds in other activities like anticancer, anti-inflammatory, immunomodulatory.

5.      It’s better to give more examples about antagonistic effects.

6.      Correct  typo errors:

Line 151. Cannabis sativa

Line 396. In vitro

Author Response

Dear reviewer, thank you for the comments and suggestions. We have changed the manuscript according to your recommendations where possible. You will find the list of changes below.

  1. In the table 1-3, it’s better to arrange the activity and plant species in each activity in alphabet order. It’s important to arrange the summary of research in the table so the readers can follow easily.

All the tables were rearanged according to the suggestion. Bioactivities were ordered alphabetically, and taxa, where applicable, were also organised alphabetically.

  1. Figure 1 is not clear. It’s better to re-draw the structures.

We corrected the figure to make the structures more clear.

  1. Many interaction references that authors described in the manuscript are not in the table. In my opinion, the table should provide full interaction examples.

We added all the references from the text in the table, when applicable. Some references discussed in the text could not be placed in any of the tables, due to the nature of the data presented in them.

Part 2.1. References 88, 90, 111-112 are not in the table 1.

These references are now added to the table, with the exception of ref.112, since it did not contain data that could be placed in the table.

Line 214-215. Reference 93 is not correct. Reference 93 is in the table 2.

We did not find issue with this reference. It was properly cited in the main text, next to appropriate description of the results.

Part 2.4. Reference 81, 115, 119, 120, 143 are not in the table 2.

Done. Reference 143 is in table 3, where it is more appropriate.

Line 346. Is the Reference 112  correct ? It mentioned about antimalarial activity.

Yes, we checked the reference, it is correct.

Line 372-373: the sentence “One of the drawbacks of

this approach is that it may not facilitate the identification of compounds that, on their

own, do not possess the medicinal properties, but in fact facilitate or enable the activity

of the active compound” need to rephrased. It’s difficult to understand.

Done.

Part 3.1. Reference 193, 126, 54 are not in the table 3.

Refernce 193 is now in the table 3. References 126 and 54 are in table 2.

Part3.2. Reference 97, 99, 197-201 are not in the table 3.  In the table 3, there are several in vivo studies but the authors did not describe in detailed any examples. 

References 97, 99, 197-201 are in table 1, since it was more appropriate. Table 3 deals only with synergistic effects. Not every paper presented in the tables is discussed in the main text. The focus was on diversity of activities and results. Tables are there for additional information and to guide the reader to appropriate research.

Figure 4. Marchatin A is not mentioned in the manuscript. It can be removed.

Marchintin A is mentioned in the table 1, and it is appropriate to show it as a representative of not common compound that were found to also have biological activity

Part3.3. The authors can make another table for this part, not included in the table 2.

Data for this type of combinations are very scarce, so we incorporated them in table 3. Most of the literature data looks into commercial drug-plant interactions, and most of the reviews on the topic of synergism with plant extracts also concentrate on this topic.

Reference 151. The author omitted nisin in the combination.

Nisin is now discussed in the text.

  1. Many references in the tables are not described. I understand the manuscript topic is very broad, the authors can not describe in detailed all the references but it seems the authors give many examples about interactions of extracts/compounds in the antimicrobial activity. It’s better to discuss about interactions of extracts/compounds in other activities like anticancer, anti-inflammatory, immunomodulatory.

This is due to the literature bias. During our analysis, certain types of data were simply not found. For example, mostly synergistic effect was considered, while all other types of interactions were rarely reported. This would correspond to the publishable data bias, where negative data is rarely published. Also, authors rarely studied influence of individual compounds if the mixture did not show good results. The bias also exists when studying synergistic effects. Due to the complicated nature of these studies, they are mostly done either with microorganisms, or, when studying other activities, mostly combinations with official/synthetic drugs was studied.

  1. It’s better to give more examples about antagonistic effects.

Yes, it would be. However, the availability of these data is still to scarce, due to publishing bias.

  1. Correct typo errors:

Line 151. Cannabis sativa

Done.

Line 396. In vitro

Done.

Reviewer 2 Report

Plant-based natural products have been used as a source for therapeutics since the dawn of civilization. Plants continuously interact with their environment, producing new compounds and ever-changing combinations of the existing ones. Interestingly, some of the compounds showed lower therapeutic activity in comparison to the entire extract they were isolated from. This review aims to develop an understanding and practical suggestions on interactions between natural extract and small-molecule compounds, which is expected to provide a useful reference for in designing smart therapeutic agents. Authors have made a detailed literature analysis and presented it in a good and organized manner. However, I have some suggestions and corrections on the article, that are appended below.

             Abstract is a good overview of the topic.

       What new insights into the topic does your review provide?

       After the Abstract, there is no Introduction section, there is need to add the introduction section in the review.

       There are many reviews written on “Interactions between natural products” There is a need to write some points that differentiate from other reviews in the “Introduction” section.

       Report evidence of reducing toxicity and discuss in the new section.

       In section 3, there is a need to add subsection with the headings such a Pharmacokinetic Interactions and Pharmacodynamic Interactions.

       It was found that disease resistance is less likely to occur against a combination of bioactive compounds than against single active molecules. How the interactions between natural products are helpful to increase the sensitivity and reducing resistance.

       The scientific search for medicinal plant extracts is challenging because of their huge complexity and variability. What are those challenges? Add in the section conclusion. 

Author Response

Dear reviewer, thank you for the comments and suggestions. We have changed the manuscript according to your recommendations where possible. You will find the list of changes below.

  • Abstract is a good overview of the topic.

Thank you.

  • What new insights into the topic does your review provide?

We added introduction section with the aim and new insights presented.

  • After the Abstract, there is no Introduction section, there is need to add the introduction section in the review.

We added introduction sections.

  • There are many reviews written on “Interactions between natural products” There is a need to write some points that differentiate from other reviews in the “Introduction” section.

We added this information. The main difference between this review and many others is that we focus on the interactions between compounds, irrespective of their grouping, origin or bioactivity, focusing on a wider audience then most of the reviews that cover this topic.

  • Report evidence of reducing toxicity and discuss in the new section.

We expended discussion on the reducing toxicity

  • In section 3, there is a need to add subsection with the headings such a Pharmacokinetic Interactions and Pharmacodynamic Interactions.

Pharmacokinetic interactions and Pharmacodymanic interactions are out of scope of this manuscript. We aimed to discuss the interaction only between natural compounds from single plant or mixed plant samples, while most of these interactions are focused on official drugs or derivatives of natural compounds.

  • It was found that disease resistance is less likely to occur against a combination of bioactive compounds than against single active molecules. How the interactions between natural products are helpful to increase the sensitivity and reducing resistance.

We expanded discussion on this topic.

  • The scientific search for medicinal plant extracts is challenging because of their huge complexity and variability. What are those challenges? Add in the section conclusion.

We added further information on this topic in the conclusion section

Round 2

Reviewer 1 Report

The author have modified manuscript based on the reviews' suggestion. The manuscript can be accepted for publication.

Few minor errors:

Lines 465, 486: in vivo in italic

Line 721-722: add full title of the reference 14.

Reviewer 2 Report

·       Most of the suggestions have been incorporated by the authors in the revised manuscript. Therefore, no issue with considering it for publication.